# Recent Advances in Early Diagnosis of Viruses Associated with Gastroenteritis by Biosensors

**DOI:** 10.3390/bios12070499

**Published:** 2022-07-08

**Authors:** Abouzar Babaei, Nastaran Rafiee, Behnaz Taheri, Hessamaddin Sohrabi, Ahad Mokhtarzadeh

**Affiliations:** 1Medical Microbiology Research Center, Qazvin University of Medical Science, Qazvin 59811-34197, Iran; abouzar.babaei95@gmail.com; 2Department of Virology, Faculty of Medical Sciences, Tarbiat Modares University, Tehran 14117-13116, Iran; rafiee.nastaran@gmail.com; 3Department of Medical Biotechnology, Faculty of Advanced Medical Sciences, Tabriz University of Medical Sciences, Tabriz 51666-16471, Iran; hbehnaztaheri@gmail.com; 4Department of Analytical Chemistry, Faculty of Chemistry, University of Tabriz, Tabriz 51666-16471, Iran; hesamsoh@gmail.com; 5Immunology Research Center, Tabriz University of Medical Sciences, Tabriz 51666-16471, Iran

**Keywords:** viral gastroenteritis, biosensor, virus detection

## Abstract

Gastroenteritis, as one of the main worldwide health challenges, especially in children, leads to 3–6 million deaths annually and causes nearly 20% of the total deaths of children aged ˂5 years, of which ~1.5 million gastroenteritis deaths occur in developing nations. Viruses are the main causative agent (~70%) of gastroenteritis episodes and their specific and early diagnosis via laboratory assays is very helpful for having successful antiviral therapy and reduction in infection burden. Regarding this importance, the present literature is the first review of updated improvements in the employing of different types of biosensors such as electrochemical, optical, and piezoelectric for sensitive, simple, cheap, rapid, and specific diagnosis of human gastroenteritis viruses. The Introduction section is a general discussion about the importance of viral gastroenteritis, types of viruses that cause gastroenteritis, and reasons for the combination of conventional diagnostic tests with biosensors for fast detection of viruses associated with gastroenteritis. Following the current laboratory detection tests for human gastroenteritis viruses and their limitations (with subsections: Electron Microscope (EM), Cell Culture, Immunoassay, and Molecular Techniques), structural features and significant aspects of various biosensing methods are discussed in the Biosensor section. In the next sections, basic information on viruses causing gastroenteritis and recent developments for fabrication and testing of different biosensors for each virus detection are covered, and the prospect of future developments in designing different biosensing platforms for gastroenteritis virus detection is discussed in the Conclusion and Future Directions section as well.

## 1. Introduction

Gastroenteritis, an inflammation of the gastrointestinal tract, leads to abdominal pain with nausea, vomiting, and diarrhea. These clinical signs can be accompanied by systemic manifestations including fever [1,2]. Gastrointestinal disorders are divided into acute or chronic infections that can be caused by infectious pathogens including viruses, bacteria, and parasites [3,4]. Acute gastroenteritis is a common disorder in people that leads to significant deaths worldwide [5,6]. Around 3–6 million annual deaths associated with gastroenteritis are estimated worldwide, of which 1.8 million deaths belonged to children aged ˂5 years, accounting for nearly 20% of total child deaths. More than 75% (1.46 million) of deaths occurred in the developing regions of Asia, Africa, and Latin America [4,7]. Meanwhile, according to the recent estimation of the Centers for Disease Control and Prevention (CDC), more than 350 million cases of acute gastroenteritis are identified throughout the USA each year [8]. In addition, statistics indicated that diarrhea is the main symptom of most gastrointestinal infections and is known as the second causative agent of preventable disease in infants and children (aged ˂ 5 years), with approximately 800,000 deaths per year [9,10,11]. In a recent annual publishing from the USA, nearly 180 million gastrointestinal disorders were detected, of which more than 4000 cases were dead [12]. Hence, most countries, especially industrial and developed nations, have improved the public health infrastructure to decrease the bacteria- and parasite-related gastrointestinal infections; however, the rate of viral infections has not declined in a comparable manner. In fact, national reports from some countries indicated that the frequency of viral gastroenteritis has increased over past years [2,13].

Since the 1940s, viruses had been suspected of being the main causative of gastroenteritis. This suspicion was confirmed in 1972, when the first characterization of a virus (Norwalk virus) was detected as an infectious agent for gastroenteritis in an outbreak of diarrhea [14,15]. Over time, the number of scientific investigations about the etiology of viral gastroenteritis steadily increased and their results led to the identification of more enteric viruses associated with acute gastroenteritis [16,17]. Now it is fully known that viral infections are responsible for about 70% of acute gastroenteritis cases in children throughout the world, and that these pathogens have a great effect on all age groups, especially severe complications such as diarrheal disease in children and elderly people aged > 60 years [18,19]. Annual estimations from the USA have shown that viral pathogens are responsible for 15 to 25 million gastroenteritis episodes, of which 3 to 5 million have visited the office and 200,000 have been hospitalized [8]. Different viruses, including rotavirus, norovirus, calicivirus, astrovirus, enteric adenovirus types 40 and 41, and some picornaviruses (enterovirus, echovirus, and parechovirus), are the main agents for viral acute gastroenteritis [19]. Clinical manifestations of viral acute gastroenteritis are not easily distinguishable from each other or from the bacterial and parasitic ones [20]. On the contrary, infection reduction, prevention of virus shedding, and successful antiviral therapy depend on the early detection of the virus, as broadly as possible. Hence, laboratory detection tests should be applied in the early stages of infection to make a specific diagnosis [21,22].

Generally, cell culture, electron microscope (EM), and serological and molecular techniques are routinely used for virus isolation, visualization, antigen or antibody measurement, and genome amplification diagnosis from the feces and body fluids of gastroenteritis sufferers [23,24,25]. Unfortunately, these techniques are complex, time consuming, expensive, need well-experienced technicians and instruments, and also have false positive and false negative results [26,27,28]. Therefore, clinical laboratories need sensitive, simple, cheap, and rapid techniques for specific diagnosis of gastroenteritis caused by viruses. Therefore, to find alternative techniques without the mentioned drawbacks of conventional detection approaches, there has been increasing attention given to applying the advanced and high-tech strategy of biosensors for detection of biomedical analytes, due to that biosensors are cheap, easy, fast, and have high sensitivity and accuracy [29,30,31,32]. Furthermore, since viral infections as well as virus-associated gastroenteritis are an important threat for humans around our planet, and especially in developing countries, there is an increasing requisition to detect these pathogens by advanced techniques [33,34,35]. Therefore, to overcome the restrictions of the existing diagnostic approaches [36], biosensors [37,38], as ultra-sensitive and specific minimized detection systems, were fabricated for early detection of virus-associated gastroenteritis from the stool and clinical specimens in recent years [26,39,40]. Consequently, their efforts resulted in designing several types of cheap, easy, fast, and highly sensitive biosensing platforms, such as immunoassay and nucleotide-assay-based sensors for the screening of viruses from clinical samples [36,39,41,42,43,44,45]. Hence, in this review article, we focus on the newest improvements in the biosensor field and an overview of the various types of constructed and tested biosensing platforms for viral gastroenteritis detection. In addition, we discuss the current diagnostic techniques of viral gastroenteritis and their limitations.

## 2. Current Diagnostic Strategies for Human Viruses Associated with Acute Gastroenteritis

Several types of conventional and recently emerged laboratory detection approaches are used for identifying enteric viruses in stool/body fluids. Some of these direct and indirect techniques are cell culture, electron microscope (EM), serological, polymerase chain reaction (PCR), and biosensors (Table 1) [26,46,47].

### 2.1. Electron Microscope (EM)

In the medical virology field, an electron microscope (EM) is one of the oldest tools for direct detection and counting of viral particles and has been in use since 1941. Observation of the first enteric virus under EM in 1971 was a turning point in the application of this technique for the detection of gastrointestinal viruses [27]. Afterward, collections of enteric viruses were discovered from clinical samples for over a decade. Despite the important role of EM in the diagnosis of gastrointestinal viruses in the last decades [48], the use of this technique has been very limited in recent years because of low sensitivity and being time consuming, expensive, and needing costly instruments and trained technicians [49].

### 2.2. Cell Culture

Traditionally, cell-culture-based assay, as a confirmatory and gold standard diagnosis, has been used for propagation and isolation, as well as observation of cytotoxicity and cytopathic effects (CPE) of enteric viruses. In addition, titration of enteric viruses is performed by plaque-forming units (PFU) and Median Tissue Culture Infectious Dose (TCID50) methods [50,51,52]. As summarized in Table 1, this technique required a significant amount of time and has low sensitivity. Meanwhile, this direct detection approach is often very susceptible to bacterial and fungal contamination. Additionally, it has been found that most viruses, such as diarrhea viruses, do not produce visible CPE in cell culture [26,53]. These are some of the listed disadvantages of cell culture that limit its usefulness for gastroenteritis viruses’ diagnosis in clinical laboratories. From the first isolation of the virus in cell culture in the 1960s, improvements such as the development of several human and animal cell lines and media cultures, production of transgenic cells, and cryopreservation of cells have increased the potential of cell-culture-based methods for enteric viruses’ cultivation, isolation, and titration. However, they are not capable of addressing all of the drawbacks of this diagnostic method, and there is still a requirement to confront the cultivation difficulties of enteric viruses [27].

### 2.3. Immunoassay Approaches

After cell-culture-based techniques, immunoassay methods have been expanded as strategies for the measurement of the immune components of viral infections as well as enteric viruses such as rotavirus and norovirus [54,55,56]. Antigens and antibodies serve as crucial detector elements in the various formats of this reliable and accurate diagnostic test [57]. Table 1 shows a list of the existing formats of the immunoassay tests, out of which the enzyme-linked immunosorbent assay (ELISA) is the most favored and easy technique for identification of enteric viruses. Furthermore, several modifications have recently been applied to increase the sensitivity and specificity of this commercial kit [26]. In addition to the ELISA, the latex agglutination test (LAT) is a rapid and easy pen-side method that is utilized for the diagnosis of viruses that cause gastroenteritis. To form the large agglutinations in this format, latex beads are first coated with specific antibodies of an enteric virus, and then they come into contact with the desired virus of the clinical samples [58]. Nevertheless, they have limited sensitivity, are time consuming and laborious, and also do not produce quantitative data [21,59].

### 2.4. Molecular Techniques

To increase the benefits of the mentioned older tests, molecular detection approaches have been developed and recently become very popular for direct diagnosis of enteric viruses, due to their high sensitivity and accuracy [21]. PCR (polymerase chain reaction), real-time PCR, and reverse transcription PCR (RT-PCR) are some of the commonly used techniques in laboratories throughout the world for the detection of most viral diseases (Table 1) [25,60,61]. Due to the high dynamic range action of nucleic acid amplification methods, including virus detection, genotyping, and viral load measurement, different companies have focused on commercializing molecular kits, and now they are used as the gold standard for most viral blood-borne and respiratory infections, as well as gastroenteritis ones [62,63,64,65]. To achieve the highest level of sensitivity and also reduce the experiment costs in comparison with monoplex PCR-based assays, multiplex RT-PCR and multiplex real-time RT-PCR (RT-qPCR) assays have been successfully designed and tested for simultaneous detection of rotaviruses, astrovirus, enteric adenoviruses, and enterovirus from fecal specimens [66,67]. However, the need for special instruments, more time, and pre-PCR processes including gene extraction and cDNA synthesis steps, as well as the inability to differentiate viable pathogens from dead ones, are the main disadvantages of these tests [68]. Therefore, the small and smart biosensing platforms are attractive, sensitive, easy, and low-cost devices for fast diagnosis of gastroenteritis viruses in the stool and body fluids of patients.

**Table 1 biosensors-12-00499-t001:** Summary of routine and newly emerged diagnostic approaches for gastroenteritis viruses.

Detection Test	Method	Time	Benefits	Limitations	Refs.
Electron microscope (EM)	TEM ^a^ SEM ^b^	3–10 days	Broad spectrum, a good test for direct detection and counting of viral particles	Low sensitivity, time consuming, expensive, and needing costly instruments and trained technicians	[27]
Cell culture	Conventional cell culture shell vial technique	1–4 days	High specificity, cheap, broad spectrum	Time consuming, low sensitivity, very susceptible to bacterial and fungal contamination. It is not applicable for viruses that do not produce visible CPE.	[26,53]
Immunoassay	ELISA ^c^ RIA ^d^ CA ^e^ MEIA ^f^ CLIA ^g^ FPIA ^h^ HI ^i^	30 min–4 h	Acceptable sensitivity, easily settled, need few reagents	Limited sensitivity, time consuming, laborious, and does not produce quantitative data.	[21,59]
Molecular techniques	PCR ^j^ Real-time PCR RT-PCR ^k^ DNA Microarrays LAMP ^l^ NGS ^m^	3–10 h	High sensitivity, specificity, and accuracy, high dynamic range action	Need for special instruments, more time, and pre-PCR processes; inability to differentiate viable pathogens from dead ones, and risk of contamination.	[62,63,64]
Biosensors	Electrochemical Optical Piezoelectric	3 min–2.30 h	Cheap, simple, rapid, high-level sensitivity and selectivity, reproducibility, low limit of detection, and accurate	-	[69]

^a^ Transmission electron microscopy. ^b^ Scanning electron microscopy. ^c^ Enzyme-linked immunosorbent assay. ^d^ Radioimmunoassay. ^e^ Immunochemiluminescent assay. ^f^ Micro-particle enzyme immunoassay. ^g^ Chemiluminescent immunoassay. ^h^ Fluorescence polarization immunoassay. ^i^ Hemagglutination inhibition. ^j^ Polymerase chain reaction. ^k^ Reverse transcription PCR. ^l^ Loop-mediated isothermal amplification. ^m^ Next-generation sequencing.

## 3. Investigation of Various Biosensing Platforms for Detection of Gastroenteritis Viruses

In the last decades, bio-scientists have focused on the recognition of biomarkers and analytes inside the human body, and their efforts led to the designing of the first sensitive and reliable bio-detector of blood glucose from clinical samples in 1962 [70]. With technology development, biomedical applications of sensors were extremely expanded and now they are employed for the detection of various human disorders [71]. Nevertheless, viral pathogen detection by biosensors is a new field that has attracted the attention of scientists in recent years; hence, these biorecognition methods were combined with conventional diagnostic strategies to find alternative assays for viral diseases without the mentioned drawbacks of old techniques [72,73]. Being cheap, simple, rapid, and having high-level sensitivity and selectivity are the main positive points of biosensing assays [69,74]. The biosensor term refers to an analytical tool with the ability to diagnose biological analytes, including microbial pathogens (virus, bacteria, etc.) [74], nucleic acids (RNA and DNA) [75,76], cancer biomarkers [77], enzymes, whole cells [78], cell receptors, etc., directly or indirectly [79]. The bio-receptor (biological recognition element), signal transducer, and amplifier are the three parts of a biosensor, of which interaction with bio-receptor and transducer produces a detectable signal (Figure 1). The analytical data of the generated signal are represented in quantitative or semi-quantitative format [72]. Commonly, biosensors are broadly characterized into several types based on the biological and transduction elements, and there are five categories of these bioassays: (i) electrochemical, (ii) optical, (iii) calorimetric, (iv) mass-based, and (v) thermometric biosensors, depending on the transducer type [80]. The selectivity, sensitivity, reproducibility, response to matrix interferences, and limit of detection (LOD) are the main properties of a biosensor to determine its performance, whether the bioreceptor type be whole virion, DNA, enzyme, antibody, or any protein [73,81]. Meanwhile, the latest advances of scientists in the nanotechnology field in introducing the wide range of nanomaterials for applications in biosensor structure paved the way for the fabrication of highly sensitive nanoscale diagnostic biosensors called nano-biosensors [29]. The high surface area, catalytic activities, unique size and shape, and electrochemical and physical-chemical attributes are the most highlighted features of nanoparticles, which introduce these particles as very important candidates for the immobilization process, signal production, and amplification, as well as increasing biosensor activity [82]. Until now, different kinds of metallic nanomaterials, including gold (Au), silver (Ag), and copper (Cu), and non-metallic nanomaterials such as graphene, carbon nanotubes, and graphite, have been used for the construction of ultrasensitive biosensors. Among them, gold particles are the most common and preferred option for this aim [83]. In addition to gold nanoparticles (AuNPs) being an important part of a biosensor, they are the first choice for antiviral, anticancer, and drug delivery purposes due to having special physical and chemical features [84,85].

The electrochemical, optical, and piezoelectric biosensors are the most frequently used bioassays for detection of viral diseases in the early stages of infection [29].

### 3.1. Electrochemical Biosensors

The high sensitivity, cost effectiveness, portability, and simplicity of electrochemical sensing methods make them first-choice options for the diagnosis of viral infections, such as gastroenteritis viruses [86]. In these small bioassays, chemical data of biological events including DNA hybridization, antigen–antibody complex, receptor–ligand binding, and enzymatic reactions at the biosensor surface are transformed into an analytically suitable signal (Figure 2a(i,ii)) [87,88]. The label-free (direct) and label-based (indirect) approaches are the two major ways to detect electrochemical biosensor signals. In the label-based approach, molecular species including organic dyes, quantum dots, and nanoparticles directly or via a bioreceptor element are attached to the target (nucleic acid, antigen, antibody, etc.) to facilitate detection. In the label-free approach, a biological target is detected in simple way, needing no special reporter [89,90,91]. The employed electrochemical biosensing methods for pathogen diagnosis as well as viruses can be subdivided into four lineages: amperometric, voltammetric, conductometric, and impedimetric [92,93,94].

### 3.2. Optical Biosensors

In these bioanalytical systems, biorecognition elements are integrated into the optical transducer system to measure the signal when a complex is formed between the recognition element and analyte (Figure 2b) [95,96]. Commonly, optical biosensors can be subdivided into two modes: (i) label-free mode, in which direct interaction of the tested target with the transducer produces a measurable optical signal; and (ii) label-based mode, where different types of labels such as fluorophores or chromophores are used for the optical signal which is created through the luminescent, fluorescent, or colorimetric method [95,97]. Recent considerable advances in the structure of optical sensors introduced them as one of the best types of biosensing methods for use in clinical detection [72]. The advantages of optical biosensing strategies are high sensitivity, specificity, compact size, low cost, and real-time detection. Hence, different kinds of optical biosensors such as surface plasmon resonance (SPR), optical fibers, adsorption, fluorescent, and luminescence are currently available in the literature and on the market [98]. Nevertheless, the SPR strategy is the most known and used version of optical biosensors for the diagnosis of gastroenteritis viruses in publications [99,100,101].

### 3.3. Piezoelectric Biosensors

Piezoelectric sensors are called active sensors because they do not need external energy to create an output signal [102]. These analytical devices, a type of mass-based biosensing strategy, work based on the alteration of oscillations due to a mass bound on the sensing surface [103,104]. The quartz crystal microbalance (QCM) biosensing platform is the most common fabricated and evaluated piezoelectric one, in which a thin layer of quartz crystal wafer is placed among two layers of metallic electrodes [105]. The alteration of the electric field causes elastic deformation of the crystal and subsequent induction of an acoustic wave. Since simple, inexpensive, and broadly available materials are used in the structure of piezoelectric biosensors, building it is faster and cheaper than other mentioned biosensing methods [106]. Meanwhile, the advantages of QCM, including acceptable sensitivity, low price, miniature size, and minimal reagents, make this label-free technique a popular diagnostic approach for viral pathogens as well as viruses associated with gastroenteritis [41,107].

**Figure 2 biosensors-12-00499-f002:**
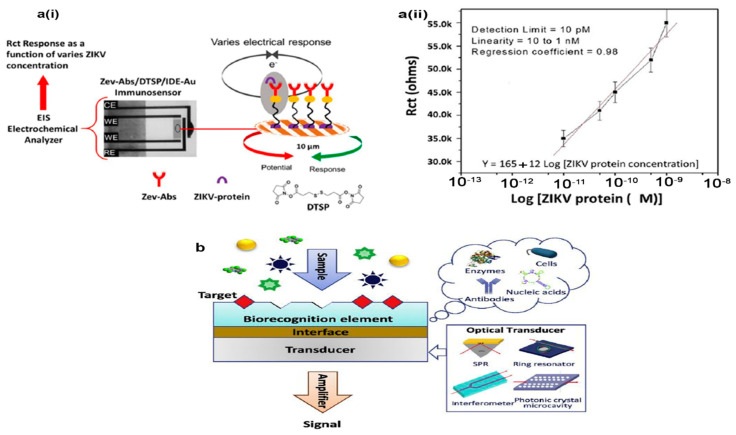
Schematics of the electrochemical (**a**(**i**) and **a**(**ii**)) and optical (**b**) sensors that are used for virus detection: **a**(**i**) picture of the impedance-based electrochemical immunosensor and **a**(**ii**); Dependence of the charge transfer resistance on the virus concentration from 10 pM to 1 nM. (**b**) Schematic diagram of optical biosensor constitution and its components. The figures were reprinted with permission from refs [96,108], Copyright 2019, Elsevier and Copyright 2020, Elsevier.

## 4. Rotavirus

Among the diarrheal disease agents, human rotavirus, a double-stranded RNA virus, is the most dominant and lethal cause of severe acute gastroenteritis in children under the age of 5, with more than 500,000 deaths globally. Older children and adults are also susceptible to rotavirus infection, but it is mostly limited to mild or moderate illness [109]. The leading transmission route of rotavirus is through fecal–oral contact; however, according to considerable shedding of rotavirus in stool during diarrhea, in some developing countries with poor sanitation systems, the virus can transmit via contaminated water. Thus, the detection of rotavirus in water sources is considered to be a preventive tool and is as important as a laboratory diagnosis that detects the virus in infected individuals [110]. Standard rotavirus detection techniques, except for the enzyme-linked immunoassay (ELISA), are not recommended by the World Health Organization (WHO) due to their laborious and resource-intensive procedure [111], and this is where various types of biosensors can be used as the potential alternative methods.

Graphene oxide (GO)-based biosensors have been applied for detecting rotavirus with various sensing mechanisms including fluorescent-based and electrochemical biosensors [112,113]. In the former sensing method, spotted GO is bound to an amino-modified glass through the difference in electric charge, then immobilized antibodies on the surface of the GO react with rotavirus for inducing fluorescence resonance energy transfer (FRET). Gold nanoparticles (AuNPs) act as a quencher and bind to rotavirus via an Ab-DNA-AuNPs complex, in which DNA has a mediator role for reducing the distance between the GO surface and AuNPs. This GO-based immune sensor has more than 15-fold quenching intensity in comparison to poliovirus and variola virus, showing a great specificity, and also its limit of detection (LOD) approximates that of conventional ELISA (10^5^ pfu mL^−1^) [112]. A comparable result with 30.7% of sensitivity has been achieved by a multi-layered GO-film-based electrochemical biosensor. This reduced GO (RGO) film is used as a working electrode and is functionalized by non-covalently binding to 1-Pyrenebutyric acid N-hydroxysuccinimide ester (PSE) molecules which also bind to and immobilize rotavirus-specific antibodies. The antibodies react specifically with rotaviruses and the process is monitored by cyclic voltammetry (CV) [113].

To distinguish various strains of rotavirus, surface-enhanced Raman spectroscopy (SERS), with the association of the chemometric analysis method, appears to be a sensitive and specific technique with an LOD of 10^4^ ffu/mL. In this approach, silver nanorod arrays are fabricated by the oblique angle vapor deposition (OAD) procedure and utilized as SERS substrates. With 100% sensitivity and >99% specificity, this SERS assay can differentiate eight different strains of rotavirus in less than a minute. Additionally, the outcome of two genotype classifications according to variations in two rotavirus capsid proteins, VP7 and VP4, indicates >96% accuracy [114]. One of the other sensitive and specific biosensors has been conducted based on genome amplification, with an LOD of 10 rotavirus nucleic acid copies per µL. This probe-based isothermal amplification method surpasses the others by using a recombinant DNA polymerase, FEN1-Bst, that not only synthesizes DNA by strand displacement but also performs the probe–primer cleavage to release the fluorescent reporter. The primers used in this system consist of a flap-probe primer complex in which the flap sequence does not bind to the template to establish a hanging state and a reverse primer. The FEN1-Bst enzyme is constructed by the binding of the flap endonuclease 1 protein (FEN1), which performs the removal of the hanging DNA sequence at the end of the polymerization, as well as Bst DNA polymerase. The fluorescence detachment leads to the excitement of fluorophores and signal amplification, which is detected in real-time analysis by amplified curves (Figure 3). When the result of rotavirus detection in stool samples was compared to the RT-PCR method, this novel technique showed 100% specificity and sensitivity, showing great potential in viral detection [115].

Fabricating a micropatterned one helps to control the shape and size of an RGO to increase its electronic function (Figure 4a). This micropatterned RGO (MRGO), when used as the basis of a field-effect transistor (FET) device, demonstrates an LOD of 10^3^ pfu/mL, which is higher than conventional ELISA in rotavirus detection. As described earlier, PSE molecules, for antibody immobilization, are attached to MRGO. Moreover, the MRGO-FET device is treated with a 1-dodecanethiol (DDT) solution to block gold electrodes from reacting to non-specific molecules. To verify MRGO-FET sensitivity, the real rotavirus-infected fecal sample was detected with the device and showed comparable results [116]. In an attempt to create a cheap, disposable rotavirus-detecting device, an Au-NP-based lateral flow immunoassay biosensor is combined with an optoelectronic device in order to provide light. In this system, phosphorescent green emitting organic light emitting diodes (OLED) and an organic photodiode (OPD) are utilized for emission and detection of light, respectively. The hybridization of the rotavirus antigen with immobilized antibodies on the test line, and consequent conjugation of them to AuNPs, results in the emission of light which decreases as AuNPs accumulate on the test line. These changes are detected by OPD and are shown on the test line, with a control line for test validation [117].

Detection of rotaviruses is not limited only to human-infecting ones. For instance, a photonic crystal biosensor has shown great sensitivity to 36 virus focus forming units (FFU) with the detection of porcine rotavirus in half an hour. Specific interaction of rotaviruses in a water sample with antibodies of the biosensor results in light reflection which is measured through a spectrometer [81]. In this review, we focus on the biosensors that have been applied for detection of human viruses cause gastroenteritis; however exemplifying non-human studies can lead to the perception of different biosensing mechanisms which can be applied for human-infecting viruses. Other types of biosensors, such as the interferometric optical detection method, electrochemical impedance spectroscopy (EIS), microstrip antenna biosensor, and fluorescence biosensor, detect rotavirus with varied sensing methods [118,119,120,121], described in Table 2.

## 5. Enteroviruses

With a positive-sense single-stranded RNA (ssRNA), the genus enterovirus belongs to the family of picornaviridae, consisting of 12 species of which only certain serotypes of echovirus and enterovirus are responsible for gastrointestinal diseases. As it is apparent from the term “entero”, these viruses are transmitted via the fecal–oral route and initially replicate in the intestinal cells, resulting in acute or persistent diarrhea. If there is a second site of infection, enteroviruses will spread there and cause other sets of symptoms according to the target organ. Due to the multiplicity of enterovirus serotypes, developing a preventive, effective vaccine is a challenging process [124,125]. This raises the necessity for rapid and precise detection of various enterovirus serotypes and their recombinant forms to supply ample time for health care providers to implement specific treatment and outbreak controlling measures [126]. Similar to rotavirus, recreational water and human sewage have been proved to be the source of contamination for both enterovirus and echovirus. The development of biosensors greatly helps to detect water and food-borne pathogens in a highly sensitive and feasible manner that can be an alternative to time-consuming conventional methods [127].

One of the laboratory efforts for detecting enteroviruses was designing a biosensor, based on gold-aptamer nanoparticles that can diagnose the targeted nucleic acid [128]. Aptamers are small single-stranded synthesized DNA or RNA oligonucleotides, and their 3D structure makes them specifically bind to their complementary nucleic acid [129]. Hybridization of aptamers and enterovirus RNA convert the purple aggregated gold nanoparticles into the red disaggregated form, which produce a signal transduction pathway that can be identified by colorimetric (color change from purple to red), spectroscopic (wavelength shift from 544 nm to 524 nm), and lateral flow (disaggregated biotin-functionalized gold nano-constructs move upon lateral flow membranes and bind to streptavidin) assays (Figure 4b). Lateral flow assay demonstrates the results in less than a minute, and all three detection techniques have a limit of detection of ≥1 × 10^−7^ M. Moreover, a desktop electronic detector device was designed for easy interpretation of the lateral flow assay for health care providers and patients, which shows the virus negativity or positivity on a screen [128,130]. In general, on account of low resistance to environmental inhibitors, low LOD, and time-saving features, aptamer-based biosensors are greatly auspicious for the detection of gastroenteritis viruses in contaminated water [131].

Oligonucleotides have been employed in label-free voltammetric methods and EIS biosensors as well as colorimetric, spectroscopic, and lateral flow sensors, but in the thiol-modified form. By using gold electrodes, the potential measured by a cyclic voltammogram demonstrated a shift from 5.18 to 122 mV for unmodified and modified gold electrodes, respectively. Additionally, the limit of detection ranged from 1 to 10 ^−1^ µM in varied concentration of complementary DNA through EIS assessment [132]. In addition to modification by oligonucleotides, gold nanoelectrodes were modified by thiol to improve the molecular self-assembly of electrode monolayers. In this label-free electrochemical biosensing method, enterovirus-specific antibodies are immobilized on gold microelectrodes, and then the electrical properties of attached enterovirus on antibodies are analyzed by EIS, which shows a limit of detection of 1.4 viral particles (VP) per mL [133].

A comparison between a commercial eSensor respiratory viral panel (eSensor RVP) and real-time PCR indicated more sensitivity and specificity of the eSensor in enterovirus detection; however, the false positivity due to cross-reactivity among other viruses’ nucleic acids is possible [134]. Enterovirus detection is also conducted by monitoring the infected cells through the virus protease activity. In this detection method, cell lines are engineered to express enterovirus protease 3C (3C^pro^) consisting of green and red fluorescent protein 2 (GFP^2^ and DsRed2, respectively). Then, following the infection of enterovirus, a FRET biosensor indicated the shift of fluorescent discharge from 600 nm (red) to 510 nm (green), measured by a fluorescent microscope and fluorometer device. The results from FRET assays resemble those of the plaque reduction assay (PRA) and dye reduction assays (DRA), with the superiority of FRET in streamlining the process over PRA and DRA. This biosensing assay has approximate specificity for other enteroviruses including echovirus but does not respond to non-picornaviruses such as human herpesvirus 1 (HSV-1). In addition, the FRET biosensor is adapted to be applied as an antiviral susceptibility detector via measuring the reduction in FRET in the existence of an anti-3C^pro^, rupintrivir [135]. Other members of the enterovirus genus have been evaluated for detection by various biosensors, but the experiments were confined to viruses that had symptoms other than gastrointestinal, such as enterovirus 71.

## 6. Norovirus

The Caliciviridae are a family of small, round-structured, non-enveloped, positive-sense single-stranded RNA viruses, which contain five genera: Norovirus, Sapovirus, Lagovirus, Nebovirus, and Vesivirus [136]. Noroviruses (NoVs) are the leading cause of infectious gastrointestinal disease (gastroenteritis) in all age groups, estimated to be associated with 200,000 deaths annually of children under 5 years old in the developing world [137]. Norovirus spreads via the fecal–oral route, person-to-person contact, and contaminated water or food [138]. Noroviruses are non-cultivatable, and no animal model has been created for human norovirus infection, so a licensed norovirus vaccine is not yet available [139]. Due to this and the highly infectious feature of the virus, the only effective way to alleviate norovirus outbreaks is through prevention and rapid detection of the virus [140]. Despite the availability of molecular, serological, and electronic microscopy techniques for viral disease detection [141,142,143], reverse transcription PCR (RT-PCR) is the only recommended and preferred standard test in clinical laboratories for norovirus RNA detection from stool and blood samples of gastroenteritis sufferers, even with its drawbacks [144]. Hence, biosensors have been considered as a promising approach in the field of norovirus detection [41]. For example, in a study conducted by Baek et al., a highly selective and sensitive electrochemical biosensor was developed for the detection of clinical HuNoV GII.4 samples using NoroBP-nonFoul (FlexL) 2 coated gold electrode sensor. The device consists of the separate assembly of eight novel peptides on the gold screen-printed electrode (Figure 4c). Among the utilized peptides, NoroBP peptides showed the highest binding affinity to noroviruses. The limit of detection of the assay was reported to be 1.7 copies/mL, which is 3-fold lower than other reported methods [145].

**Figure 4 biosensors-12-00499-f004:**
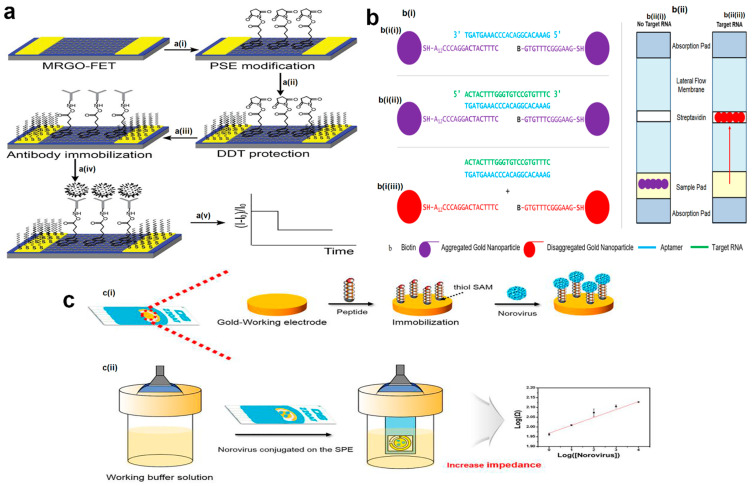
(**a**) Real-time detection of rotavirus antibodies using a micropatterned reduced graphene-oxide-based field-effect transistor. **a**(**i**) Connection of PSE on MRGO surface, **a**(**ii**) DDT treatment, **a**(**iii**) immobilization of rotavirus-specific antibodies, **a**(**iv**) rotavirus capture, and **a**(**v**) real-time monitoring of rotavirus by electrical signaling. (**b**) Schematic illustration of the gold-aptamer-nanoparticle-based biosensor with (**b**(**i**)) colorimetric and (**b**(**ii**)) lateral flow response. (**b**(**i**(**i**))) Nanoparticles are functionalized with Sequence A (SH-5′ A12CCC AGG ACT AC T TTC 3′) and Sequence B (biotin -5′ GTG TTT CGG GAA G 3′ -SH) then aggregate by aggregating oligonucleotide (aptamer). (**b**(**i**(**ii**))) Oligonucleotide hybridizes with target nucleic acid (conserved enterovirus sequence) which (**b**(**i**(**iii**))) makes the nanoparticles, disaggregates, and changes their colors into red. (**b**(**ii**(**i**))) Aggregated nanoparticles are not able to move up through the lateral flow membranes, while (**b**(**ii**(**ii**))) in disaggregated form, they flow via lateral flow and bind to streptavidin due to biotin functionalization. (**c**) Norovirus detection by use of an impedance electrochemical biosensor. (**c**(**i**)) Immobilization of peptide self-assembly monolayers (SAMs) on the Au-working electrode. (**c**(**ii**)) Affinity strength of dropped norovirus, conjugated with the peptide on the gold screen-printed electrode (SPE), is measured using electrochemical impedance spectroscopy analysis. The figures were reprinted with permission from refs. [116,128,145], Copyright 2013, Elsevier; Copyright 20211, Multidisciplinary Digital Publishing Institute (MDPI); and Copyright 2018, Elsevier, respectively.

Meanwhile, it has been shown that conjugation of concanavalin A as a NoV detection element with a nanostructured gold electrode can lead to the production of a sensitive and selective electrochemical biosensor (Figure 5). A linear relationship between the current and concentration of NoV (in the range of 10^2^ and 10^6^ copies/mL) was estimated to be (R2 = 0.998) for the designed biosensor. The limit of detection of the assay was reported to be 35 copies/mL with a selectivity of approximately 98% and a short time (1 h) needed to perform it [146].

One of the precise and highly specific biosensor-based detection methods is to measure the pathogens via their genome. In this regard, an electrochemical biosensor is designed and operates based on electrical resistance changes of nanoparticle-modified carbon nanotubes, which is determined by the concentration of hybridized target norovirus DNA with probe DNA and shows an LOD of 8.8 pM [147]. In another recent study, the CdSe–ZnO flower-rod core-shell structure (CSZFRs) photoelectrochemical (PEC)-based biosensor was constructed for detection of norovirus RNA (NoV RNA) (Figure 6a). The ion-exchange method was used for CSZFR preparation and the assay benefits from advantages such as speed, simplicity, and convenience. The limit of detection of the assay was reported to be 0.50 nM, with a good linear relationship between the photocurrent and the NoV RNA density in the range of 0–5.10 nM [148].

In addition, of the electrochemical systems, the SPR biosensing method is the most common type of optical sensor for the detection of noroviruses. In a recent study, a V-shaped trench biosensor was designed to detect norovirus-like particles GII.4 by use of the SPR method. As depicted in Figure 6b, a sandwich assay was conducted by applying a quantum dot fluorescent dye and monoclonal antibody as the label and biorecognition molecule, respectively. The limit of detection of the assay was 0.01 ng/mL, which was estimated to be equal to 100 virus-like particles [149]. While in SPR biosensors, the characteristics of gold nanoparticles highly influence the fluorescence signals, in different localized SPR (LSPR) biosensors, fluorescent quantum dots (QDs) act as fluorescent reporters. To develop an LSPR fluorescence-based biosensor for detection of norovirus through antigen–antibody interaction, the QDs, AuNPs, and antibodies are hybridized. After bonding of the Ab-QD-AuNP complex to the virus, the initially quenched fluorescent signals will be recovered via generated steric hindrance LSPR signals between two nanoparticles, which is dependent on analyte concentration. The detection limit of this method was found to be 95.0 copies of isolated norovirus per mL, showing a hundred-fold higher LOD than commercial ELISA [150].

In addition, recently a real-time quartz crystal microbalance version of piezoelectric biosensors has been optimized to recognize, capture, and amplify NoV by Selvaratnam et al. The studied biosensor applied the Padlock probe and rolling circle amplification (RCA) for the detection of the viral genome [151]. In addition to the beforementioned biosensing strategies that were fabricated for NoV monitoring, a summary of the electrochemical, optical, and piezoelectric studied biosensors for the detection of NoV is listed in Table 3.

## 7. Enteric Adenovirus 40 and 41

Adenoviruses are small, non-enveloped, linear double-stranded DNA viruses, classified under the viral family adenoviridae, which contains more than 60 different identified serotypes [155]. Serotypes 40 and 41 are known as enteric adenoviruses, which have been estimated to be the cause of the 5–20% of hospitalizations for diarrhea in developed countries [2]. Adenoviruses are transmitted via the fecal–oral route, contaminated water, and fomites. The virus remains on surfaces for a long time and is resistant to surfactant-based disinfectant solvents and temperatures [156]. Routinely, PCR assay, a more sensitive and rapid diagnostic test, is preferred over virus isolation in cell cultures and antibody-based immunoassays such as immunochromatography, immunofiltration, and direct immunofluorescence methods; however, it also takes several hours to obtain a result, and has technical complexity [157,158,159]. Therefore, the development of sensitive and rapid assays which can detect the virus at the early stage of infection when the copy number of the virus is low is essential to control the spread of the virus. Recently, several types of biosensing strategies, especially electrochemical versions, have been evaluated for diagnosing different serotypes of adenoviruses in clinical samples, while there is only one published for enteric adenovirus 40 and 41 [160,161]. There is a label-free nucleic-acid-based electrochemical sensor that was developed by Song et al. for diagnosing the human adenovirus 40/41 fiber gene [162]. To construct this sensor, polypyrrole (PPy) film was electropolymerized on nanoporous silicon (NPS), and then a 25 bp DNA probe (PDNA), which targets fiber genes, was fixed on the PPy-coated NPS. The conductivity change, caused by the fixed PDNA and hybridized target DNA (tDNA), was expressed as a normalization factor (γ). The sensitivity of the assay for detecting target DNA was −1.54 µM^−1^ in the linear range of 0.4 to 1.0 µM of tDNA.

## 8. Conclusions and Future Direction

With contributing to 70% of all gastrointestinal infections, viruses have been recognized as the leading cause of diarrheal diseases and gastroenteritis globally, which in some cases leads to death and hospitalization of children <5 years old, as well as adults. Due to the fecal–oral transmission route of gastrointestinal viruses, the infection can easily disseminate among a society, mostly through direct or indirect exposure to pathogen-loaded feces, food, and water. This condition aggravates in developing countries, where the unavailability of clean water and sanitation leads to the prevalence of food- and water-borne diseases. Mortality and hospitalization rates, as well as the burden of chronic complications and possibility of reinfection carried by infected individuals, call for a prompt diagnosis along with prevention of the infectious agent to further spreading. Conventional methods, regardless of their limitations in sensitivity, expedition, and specificity, have been greatly developed for routine diagnosis; however, they have never acted as a preventive technique to detect viral pathogens in contaminated resources. Moreover, for some viruses, such as rotavirus, the routine detection technique is only confined to one method recommended by the WHO due to the laborious procedure of other techniques that demand well-trained personnel.

Since the mentioned obstacles necessitate a novel substitute method, recent experiments attempt to employ biosensors in viral detection which not only make up the deficiencies of conventional methods through their advantages, but also help to detect viral pathogens in infected water. Rapid, cheap, and simple properties of biosensors, plus their high sensitivity and selectivity, make this unconventional system a promising approach to overcome existing challenges. They are even capable of distinguishing different serotypes of viruses and are also able to detect them in much lower quantities, defined as the limit of detection, compared to conventional methods. Among various biosensors discussed in this review, label-free methods, whether it be an electrochemical or optical one, are of more interest, as they do not need any reporter for the target molecule. In addition, of the nanomaterials used in biosensor structure, gold nanoparticles have been utilized frequently in different studies, showing their supremacy.

The evolution of nanotechnology has enhanced the diagnosis of viruses to such an extent that not even virus particles but also virus-infected cells can be detected via modern technology such as atomic force microscopy [163]. The latest studies have propelled towards the development of microfluidic and smartphone-based biosensors which have demonstrated great efficiency for detecting different viruses, including the dengue virus and hepatitis B virus [73,164]. Moreover, the detection of one virus particle, such as the influenza A virus, is possible by an antibody-modified nanowire FET [165]; however, they have not been applied for gastroenteritis-related viruses yet. Even though there are several studies on the effectiveness of biosensors in gastroenteritis-related virus detection, the multiplicity of viruses that cause gastrointestinal diseases demands more research on viruses that have not been assessed yet. In addition to detecting purposes, future studies might include genotyping of different strains of viruses. To sum up, further investigations must be conducted to ensure the usage of biosensors before they can be approved by health organizations as regular tests in laboratories.

## Figures and Tables

**Figure 1 biosensors-12-00499-f001:**
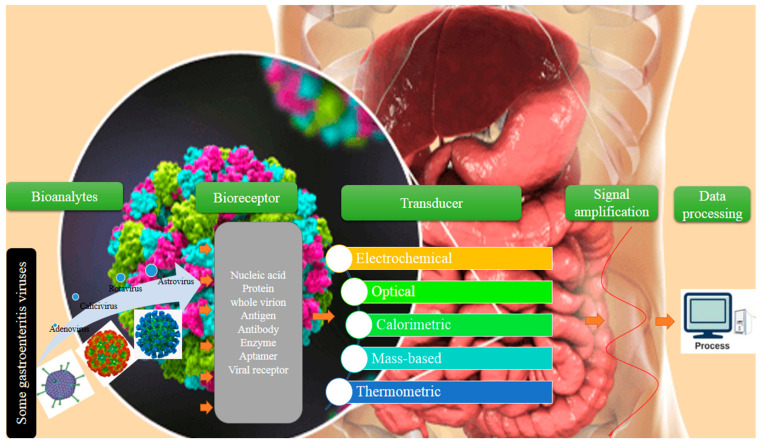
A schematic presentation of biosensor components applied for diagnosis of viruses associated with gastroenteritis. Biosensors are classified into electrochemical, optical, calorimetric, mass-based, and thermometric based on the types of their transducers and bioreceptors. The interaction of bioanalytes such as gastroenteritis viruses (rotavirus, calicivirus, astrovirus, and adenovirus) with different types of bioreceptor components including nucleic acid, protein, whole virus, antigen, etc., in various biosensing platforms produce a measurable signal through the transducer. Eventually, the signal is quantified using an analyzing system.

**Figure 3 biosensors-12-00499-f003:**
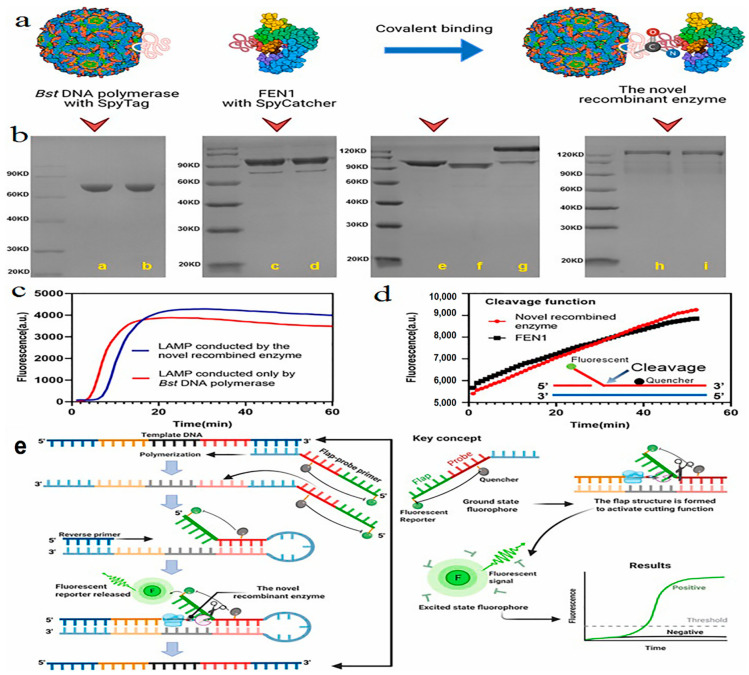
The recombinant FEN1-*Bst* DNA polymerase and a schematic illustration of flap-probe-based isothermal nucleic acid amplification method for rotavirus genome detection. (**a**) Schematic presentation of recombination through covalent binding of the SpyTag/SpyCatcher system. (**b**) SDS-PAGE confirmation of *Bst* DNA polymerase with SpyTag, FEN1 with SpyCatcher, and the new recombinant enzyme: lane a and b are reduced and non-reduced *Bst* DNA polymerase (SpyTag) protein, respectively; lane c is reduced FEN1 (SpyCatcher) protein while lane d is non-reduced one; *Bst* DNA polymerase (SpyTag) in lane e and FEN1 (SpyCatcher) in lane f form the novel recombinant enzyme protein (lane g); and lane h and i indicate novel recombinant enzyme protein. (**c**) Common LAMP reaction utilizing the novel recombinant enzyme. (**d**) Cleavage reaction of the flap structure using the novel recombinant enzyme to release fluorescent reporter that can be detected in real-time. (**e**) Overview of the mechanism and key concepts such as polymerization, flap structure creation, cleavage to release the fluorescent reporter, and real-time detection by analyzing fluorescent signal. Notes: LAMP, loop-mediated isothermal amplification. This figure was reprinted with permission from ref [115] Copyright 2022, Elsevier.

**Figure 5 biosensors-12-00499-f005:**
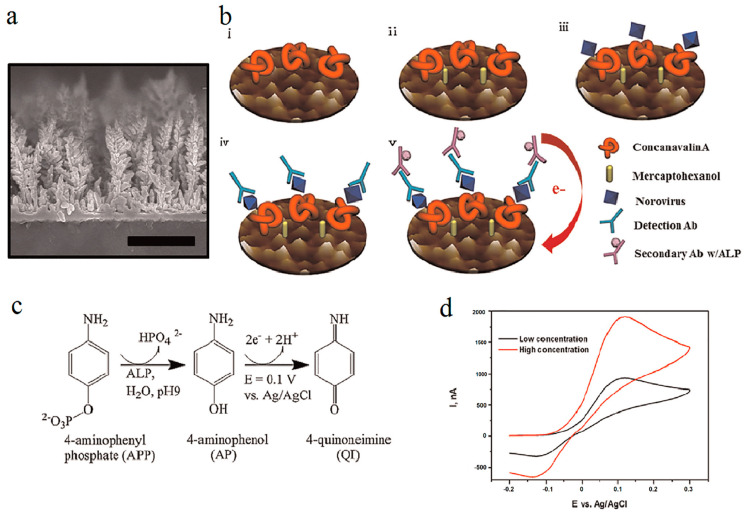
Preparation and characterization of concanavalin A (ConA)-based electrochemical biosensor. (**a**) The scanning electron microscope (SEM) image. (**b**) Schematic illustration of biosensor: (**b**(**i**)) after fixation of ConA, (**b**(**ii**)) after blocking using mercaptoethanol (MCH), (**b**(**iii**)) after norovirus fixation, (**b**(**iv**)) after fixation of first Ab, and (**b**(**v**)) after fixation of secondary Ab using (**c**) ALP-labeled antibody transforms APP to AP, which is then oxidized and produces a current at the electrode that is identical to the quantity of NoV bound to the sensor surface. (**d**) Signal reading by use of cyclic voltammetry (CV). The figure was reprinted with permission from ref. [146], Copyright 2014, Elsevier.

**Figure 6 biosensors-12-00499-f006:**
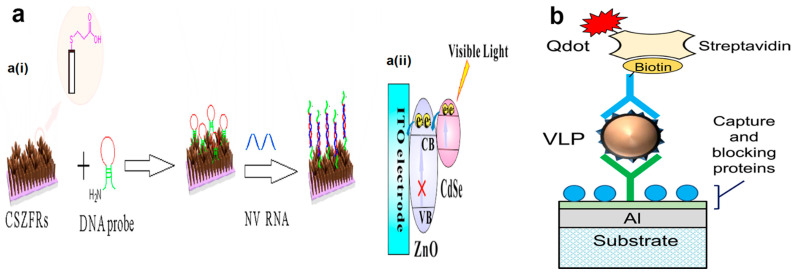
(**a**) A schematic presentation of (**a**(**i**)) the synthesis procedure and (**a**(**ii**)) electron transfer of CdSe–ZnO flower-rod core-shell structure (CSZFRs)-based photoelectrochemical (PEC) biosensor. (**b**) Schematic illustration of the surface of the V-trenches designed for detection of norovirus virus-like particle (VLP). The figures were reprinted with permission from refs. [148,149], Copyright 2018, Multidisciplinary Digital Publishing Institute (MDPI) and copyright 2017, Elsevier, respectively.

**Table 2 biosensors-12-00499-t002:** The list of various types of biosensors used for rotavirus detection.

Sensing Technique	Matrix Material	Sample	Assay Duration	Linear Range	LOD	Ref.
Electrochemical impedance spectroscopy	AuNPs	Rotavirus vaccine antigen	55 min	4.6–4.6 × 10^4^ pfu/mL	2.3 pfu/mL	[122]
A label-free field-effect transistor (FET)	Reduced graphene oxide	Rotavirus antigen solution	__	10^1^–10^6^ particle/mL	__	[123]
Microstrip antenna biosensor	Microstrip antenna	Blood serum	__	__	__	[118]
Optical	3D nanoporous photonic crystal	Commercial rotavirus antigen	2 h	6.35 µg/mL–1.27 mg/mL	__	[121]
Optical interferometric	Interferometer	Commercial rotavirus solution	__	__	1.37 ng/mL	[120]
Optical fluorescent	Well glass slide	1. Stool 2. Infected cell culture	3 min	__	1. 1 × 10^5^ pfu/mL 2. 1 × 10^4^ pfu/mL	[119]

Abbreviations: Gold nanoparticles (AuNPs); label-free field-effect transistor (FET); 3-dimensional nanoporous (3D nanoporous).

**Table 3 biosensors-12-00499-t003:** The list of various biosensors used for norovirus detection.

Sensing Technique	Matrix Material	Sample	Assay Duration	Linear Range	LOD	Ref.
SPR	Polyacrylate beads	Strawberries and milk	15 min	N/A	up to 10 units/mL	[152]
SERS- ICG	Colloidal gold	Centrifuged fecal specimen	~15 min	3~150 ng /mL	0.5 ng/mL	[153]
Plasmonic biosensor	AuNP	NoV capsid protein and HuNoV	-	10~10^5^ copies/mL	0.1 ng/mL NoV and 9.9 copies/mL HuNoV	[101]
Naked-eye biosensor	Polyhedral Cu nanoshell deposited AuNP	Stool	10 min	2.7 × 10^3^~2.7 × 10^5^ copies	2700 copies	[154]

Abbreviations: Human norovirus (HuNoV); surface plasmon resonance (SPR); surface-enhanced Raman spectroscopy (SERS); immunochromatography (ICG); gold (AU); carbon nanotubes (CNT); norovirus-like particles (NoV VLPs); magnetic nanoparticle (MNP); gold nanoparticle (AuNP); quantum dots (QD).

## Data Availability

Not applicable.

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
