# Peer review of "Recent Advances in Early Diagnosis of Viruses Associated with Gastroenteritis by Biosensors"

_biosensors, 2022, doi:10.3390/bios12070499_

Round 1

Reviewer 1 Report

Viruses are the main causative agent (~70%) of gastroenteritis episodes and their specific and early diagnosis via laboratory assays is very helpful to having successful antiviral therapy and reduction of infection burden. Therefore the topic chosen in this manuscript was valuable. However author did not control the connotation adequately in the terms of representative and comprehensive aspects, and the content provided in the manuscript was lacking organization, it was recommended to adjust the structure appropriately. At the same time, the English expression needs to be greatly improved. I think the manuscript in its current form is not suitable for publication on the biosensors.

Author Response

#Reviewer 1:

Comments and Suggestions for Authors

1) Viruses are the main causative agent (~70%) of gastroenteritis episodes and their specific and early diagnosis via laboratory assays is very helpful to having successful antiviral therapy and reduction of infection burden. Therefore, the topic chosen in this manuscript was valuable. However, author did not control the connotation adequately in the terms of representative and comprehensive aspects, and the content provided in the manuscript was lacking organization, it was recommended to adjust the structure appropriately. At the same time, the English expression needs to be greatly improved. I think the manuscript in its current form is not suitable for publication on the biosensors.

Response: Thank you for your consideration. Due to relatively low number of studies, regarding the biosensing method for detection of gastroenteritis viruses, we have tried our best to conclude all of the related articles in this review. First of all, we have discussed the advantages and disadvantages of the conventional techniques for detecting of gastroenteritis-associated viruses, then, we have given detailed information about the novel biosensing strategies and their different types which have been classified into three categories of electrochemical, optical, and piezoelectric biosensors. For each gastroenteritis-caused virus, we have systematically described the structure, pathogenesis, and epidemiology of its and the designed biosensors for that specific virus following other examples of studied biosensors, that have been presented in a word table. As well, to develop and enhance the excellence of the article, all authors reviewed and improved the whole of the manuscript and the final version of the paper has been reviewed by a native English speaker. All the improvements are marked up using "Track changes" function and highlighted in green. We hope that they will be satisfactory.

Dear reviewer, if you are considering a particular structure for categorizing information in this article, we will be appreciated to have your opinion in a clearer explanation so that we can restructure the review, if any modification is possible.

Reviewer 2 Report

In the manuscript entitled “Recent advances in early diagnosis of viruses associated with gastroenteritis by biosensors”, Babaei et al. review modern methods employed for detection of viral infections in humans, focusing on gastroenteritis.

Despite the manuscript is well organized and written in a clear language, I recommend the authors to perform a revision. My comments are listed below.

My major comment is related to the nanotechnology-based approaches noted by the authors. Indeed, this is a very important point. In fact, with respect to the detection of viruses, nanotechnology-based methods allow one to reliably detect even single viral particles: in this way, Patolsky et al. demonstrated the detection of single Influenza A virus particles with a nanowire biosensor [Patolsky F. et al., Electrical detection of single viruses. Proc. Natl. Acad. Sci. USA, 2004, 101 (39) 14017-14022, https://doi.org/10.1073/pnas.0406159101]. The authors only mention another review by Mokhtarzadeh in this respect (Ref. 27). I recommend the authors to reflect the application of nanotechnology-based methods (including atomic force microscopy and nanowire biosensors) for the early revelation of virus infections in humans in more detail.

Minor points:

1. Figure 2a, right panel: something seems to be wrong with the X scale or the X axis title in this panel. The 10 pM detection limit is mentioned, but 10-11 pM is equal to 10-23 M. Accordingly, M is expected in the X axis title instead of pM. Please, revise this issue.

2. I recommend the authors to check the manuscript for misprints, and to correct formatting issues. The lines are not numbered in the PDF file, so I only give below several points, which seem to be confusing:

Section 3.2, first sentence: Expected: “In these bioanalytical systems …”

P. 16, last paragraph: Too long sentence. Expected: “Detection of rotaviruses is not only limited to human-infecting ones. For instance, …”

P. 19, last paragraph: Expected: “… as well as in colorimetric, …”. Furthermore: the sentence “Besides modification of oligonucleotide, gold nanoelectrodes have been altered by thiol to improve molecular self-assemblies of electrode monolayers” seems to be unclear. Do I understand correctly that besides modification by oligonucleotides, gold nanoelectrodes were modified by thiol to improve molecular self-assembly of electrode monolayers?

In addition, last five pages should be eliminated (manuscript preparation instructions were not deleted).

The manuscript can be published after the authors address these comments.

Sincerely,

The reviewer

Author Response

#Reviewer 2:

Comments and Suggestions for Authors

In the manuscript entitled “Recent advances in early diagnosis of viruses associated with gastroenteritis by biosensors”, Babaei et al. review modern methods employed for detection of viral infections in humans, focusing on gastroenteritis. Despite the manuscript is well organized and written in a clear language, I recommend the authors to perform a revision. My comments are listed below.

The manuscript can be published after the authors address these comments.

Thank you for your valuable comments. We have applied all of your valuable suggestions accordingly and highlighted the applied changes are marked up using "Track changes" function as well as highlighted in green.

1) My major comment is related to the nanotechnology-based approaches noted by the authors. Indeed, this is a very important point. In fact, with respect to the detection of viruses, nanotechnology-based methods allow one to reliably detect even single viral particles: in this way, Patolsky et al. demonstrated the detection of single Influenza A virus particles with a nanowire biosensor [Patolsky F. et al., Electrical detection of single viruses. Proc. Natl. Acad. Sci. USA, 2004, 101 (39) 14017-14022, https://doi.org/10.1073/pnas.0406159101]. The authors only mention another review by Mokhtarzadeh in this respect (Ref. 27). I recommend the authors to reflect the application of nanotechnology-based methods (including atomic force microscopy and nanowire biosensors) for the early revelation of virus infections in humans in more detail.

Response: Thank you for your great remark. Since we tried to focus on gastroenteritis viruses in this review, we have added the information of two studies about atomic force microscopy and nanowire biosensor in future direction section which have the reference number 142 and 144, respectively. Evolution of nanotechnology has enhanced the diagnosis of viruses to such an extent that not even virus particles but also virus-infected cells can be detected via modern technology like atomic force microscopy (Kuznetsov, Victoria et al. 2003). Latest studies have propelled towards the development of microfluidic and smartphone-based biosensors which have demonstrated great efficiency for detecting different viruses including dengue virus and hepatitis B virus (Hassanpour, Baradaran et al. 2018, Eivazzadeh-Keihan, Pashazadeh-Panahi et al. 2019). Moreover, the detection of one virus particle, like influenza A virus, is possible by an antibody-modified nanowire FET (Patolsky, Zheng et al. 2004); however, they have not been applied for gastroenteritis-related viruses yet.

Minor points:

2) Figure 2a, right panel: something seems to be wrong with the X scale or the X axis title in this panel. The 10 pM detection limit is mentioned, but 10-11 pM is equal to 10-23 M. Accordingly, M is expected in the X axis title instead of pM. Please, revise this issue.

Response: Your comment has been applied accordingly.

3) I recommend the authors to check the manuscript for misprints, and to correct formatting issues. The lines are not numbered in the PDF file, so I only give below several points, which seem to be confusing:

Response: We have applied your hint accordingly.

4) Section 3.2, first sentence: Expected: “In these bioanalytical systems …”

Response: Your comment has been applied accordingly.

5) P. 16, last paragraph: Too long sentence. Expected: “Detection of rotaviruses is not only limited to human-infecting ones. For instance, …”

Response: According your suggestion we have separate these two sentences from each other.

6) P. 19, last paragraph: Expected: “… as well as in colorimetric, …”. Furthermore: the sentence “Besides modification of oligonucleotide, gold nanoelectrodes have been altered by thiol to improve molecular self-assemblies of electrode monolayers” seems to be unclear. Do I understand correctly that besides modification by oligonucleotides, gold nanoelectrodes were modified by thiol to improve molecular self-assembly of electrode monolayers?

Response: Thank you for your hint. Yes. Your mean is correct. It was a typos error that corrected and highlighted accordingly.

7) In addition, last five pages should be eliminated (manuscript preparation instructions were not deleted).

Response: According to your comment, we have removed the manuscript preparation section from the end of the version.

Reviewer 3 Report

Mokhtarzadeh and coauthor reviewed the updated improvements in the employing of different types of biosensors such as electrochemical, optical, and piezoelectric for sensitive, simple, cheap, rapid, and specific diagnosis of human gastroenteritis viruses, which are associated with gastroenteritis, one of the main worldwide health challenges. The paper discussed the importance of viral gastroenteritis, types of viruses that cause gastroenteritis, reasons for the combination of conventional diagnostic tests, the current laboratory detection tests for human gastroenteritis viruses and their limitations, recent developments for fabrication and testing of different biosensors, and the prospect of future developments in designing different biosensing platforms for gastroenteritis virus’s detection. This review is meaningful and informative. Therefore, I recommend the paper be accepted by Biosensors after addressing the following issues in a minor revision.

1. The foot notes of Table 1 are missing. What do “a, b, c…” in Table 1 mean? The authors should clarify about it.

2. The labels of Fig. 3 are not consistent. It would be better if the authors could replace “E” with “e”.

3. The authors should double check the format of the references. For example, ref. 41 and 101.

Author Response

#Reviewer 3:

Comments and Suggestions for Authors

Mokhtarzadeh and coauthor reviewed the updated improvements in the employing of different types of biosensors such as electrochemical, optical, and piezoelectric for sensitive, simple, cheap, rapid, and specific diagnosis of human gastroenteritis viruses, which are associated with gastroenteritis, one of the main worldwide health challenges. The paper discussed the importance of viral gastroenteritis, types of viruses that cause gastroenteritis, reasons for the combination of conventional diagnostic tests, the current laboratory detection tests for human gastroenteritis viruses and their limitations, recent developments for fabrication and testing of different biosensors, and the prospect of future developments in designing different biosensing platforms for gastroenteritis virus’s detection. This review is meaningful and informative. Therefore, I recommend the paper be accepted by Biosensors after addressing the following issues in a minor revision.

Dear reviewer, thanks for your complementary comments. We have done all of your valuable comments accordingly and highlighted all of the changes in the manuscript.

1) The foot notes of Table 1 are missing. What do “a, b, c…” in Table 1 mean? The authors should clarify about it.

Response: Thank you for your hint. According your comment, the desired foot notes have been listed and highlighted below Table 1.

2) The labels of Fig. 3 are not consistent. It would be better if the authors could replace “E” with “e”.

Response: Your comment has been applied accordingly.

3) The authors should double check the format of the references. For example, ref. 41 and 101.

Response: We have applied your valuable comment and corrected the mentioned references accordingly.

Round 2

Reviewer 1 Report

The present manuscript have gained significant improvement after revision. I recommend publication to the biosensors